# Targeted transcriptomic study of the implication of central metabolic pathways in mannosylerythritol lipids biosynthesis in *Pseudozyma antarctica* T-34

**Keisuke Wada**[ID][1☯], **Hideaki Koike**[2☯], **Tatsuya Fujii**[1], **Tomotake Morita**[ID][3]*

**1** Research Institute for Sustainable Chemistry, National Institute of Advanced Industrial Science and Technology (AIST), Kagamiyama, Higashi-Hiroshima, Hiroshima, Japan, **2** Bioproduction Research Institute, National Institute of Advanced Industrial Science and Technology (AIST), Higashi, Tsukuba, Ibaraki, Japan, **3** Research Institute for Innovation in Sustainable Chemistry, National Institute of Advanced Industrial Science and Technology (AIST), Higashi, Tsukuba, Ibaraki, Japan

☯ These authors contributed equally to this work.
* morita-tomotake@aist.go.jp

**Data Availability Statement:** All gene expression files are available from the Gene Expression Omnibus (GSE47171).

## Abstract

*Pseudozyma antarctica* is a nonpathogenic phyllosphere yeast known as an excellent producer of industrial lipases and mannosylerythritol lipids (MELs), which are multi-functional glycolipids. The fungus produces a much higher amount of MELs from vegetable oil than from glucose, whereas its close relative, *Ustilago maydis* UM521, produces a lower amount of MELs from vegetable oil. In the present study, we used previous gene expression profiles measured by DNA microarray analyses after culturing on two carbon sources, glucose and soybean oil, to further characterize MEL biosynthesis in *P. antarctica* T-34. A total of 264 genes were found with induction ratios and expression intensities under oily conditions with similar tendencies to those of MEL cluster genes. Of these, 93 were categorized as metabolic genes using the Eukaryotic Orthologous Groups classification. Within this metabolic category, amino acids, carbohydrates, inorganic ions, and secondary metabolite metabolism, as well as energy production and conversion, but not lipid metabolism, were enriched. Furthermore, genes involved in central metabolic pathways, such as glycolysis and the tricarboxylic acid cycle, were highly induced in *P. antarctica* T-34 under oily conditions, whereas they were suppressed in *U. maydis* UM521. These results suggest that the central metabolism of *P. antarctica* T-34 under oily conditions contributes to its excellent oil utilization and extracellular glycolipid production.

## Introduction

*Pseudozyma antarctica* is a basidiomycetous yeast belonging to Ustilaginomycetes, which includes the corn smut fungus, *Ustilago maydis* [1,2], and is known to extracellularly produce not only lipases but also a biodegradable plastic-degrading enzyme that hydrolyzes

**Funding:** The authors received no specific funding for this work.

**Competing interests:** The authors have declared that no competing interests exist.

polybutylene succinate and polybutylene succinate-co-adipate [3]. *Pseudozyma antarctica* T-34 was isolated in Tsukuba, Japan, as a producer of extracellular glycolipids, mannosylerythritol lipids (MELs; **Fig 1**), which consist of 4-*O*-β-D-mannopyranosyl-*meso*-erythritol as the hydrophilic moiety and fatty acids as the hydrophobic moiety [4]. MELs not only have high potential as eco-friendly biosurfactants due to their excellent surface activity, but also have attracted considerable recent interest because of their unique properties, including self-assembly, anti-tumor and cell differentiation induction activities, and moisturizing and hair-repairing properties [5, 6]. Further improvements to the mass production of MELs, and their applications to life science, nanotechnology, and environmental technology, have been investigated [7–10].

*Pseudozyma antarctica* T-34 produces large amounts of MELs when grown in culture containing vegetable oil as the carbon source, and the production yield reaches 140 g/L using *n*-alkanes as the carbon source [11]. *Pseudozyma aphidis* is also an efficient producer of MELs with a yield of more than 165 g/L from vegetable oil as the main carbon source [12, 13]. It should be noted that a closely related fungus, *Ustilago maydis* UM521, produces lower amounts of MELs from vegetable oils than yeast strain of the genus *Pseudozyma*, including the strain T-34. While *U. maydis* DSM4500 was reported to produce extracellular glycolipids in the yield of 30 g/L from 45 g/L of sunflower oil as main carbon source, the glycolipids are the mixture of MELs and cellobiose lipids (CLs) [14]. The *Pseudozyma* strains therefore have considerable potential for large-scale industrial production of MELs using vegetable oil.

Recently, we reported the genome sequence of *P. antarctica* T-34 [15] and found that this yeast has an oleaginous nature based on genomic and transcriptomic analyses; a gene encoding an ATP/citrate lyase conserved in oleaginous strains found in the genome of strain T-34 [16]. Using gene set enrichment analysis, the gene sets related to fatty acid metabolism were significantly upregulated in the presence of vegetable oil, and the gene cluster for MEL biosynthesis was also highly expressed in *P. antarctica* T-34, regardless of whether the carbon source was glucose or soybean oil (**Fig 1**). Genomic analysis showed that the *P. antarctica* T-34 genome was similar to that of *U. maydis* UM521 in chromosomal organization. However, the gene sets enriched were significantly different, suggesting different regulatory mechanisms. In addition, MEL production from vegetable oil by *P. antarctica* T-34 (30 g/L of MELs, mainly MEL-A) was more effective than that of *U. maydis* UM521 (slight amounts of glycolipids on TLC analysis), suggesting that *P. antarctica* T-34 may utilize vegetable oil for growth and glycolipid synthesis. However, the characteristics of oil utilization by *P. antarctica* T-34 remained unclear [16].

In the present study, we further analyzed transcriptomic data to improve our understanding of the molecular mechanisms of oil utilization and MEL production in *P. antarctica* T-34. The transcriptional analysis focused on genes that acted similarly to the MEL biosynthesis gene cluster, and were selected using "guilt-by-association" in the induction ratio. The gene expression intensities revealed that genes related to central metabolic pathways such as glycolysis and the tricarboxylic acid cycle (TCA) were upregulated in *P. antarctica* T-34 when compared to *U. maydis* UM521 under oily conditions. These results suggest that *P. antarctica* T-34 is adapted to aerobically produce larger amounts of MELs from vegetable oil by modification of its central metabolic system. Overall, insight into the oil utilization capacity of microorganisms will lead to more effective strategies for using feedstocks to produce functional bio-based materials.

## Materials and methods

### Microorganisms

*Pseudozyma antarctica* (formerly *Candida antarctica*) T-34 used in this study was isolated as a MEL producer using soybean oil as the sole carbon source [4]. *Ustilago maydis* DSM14603

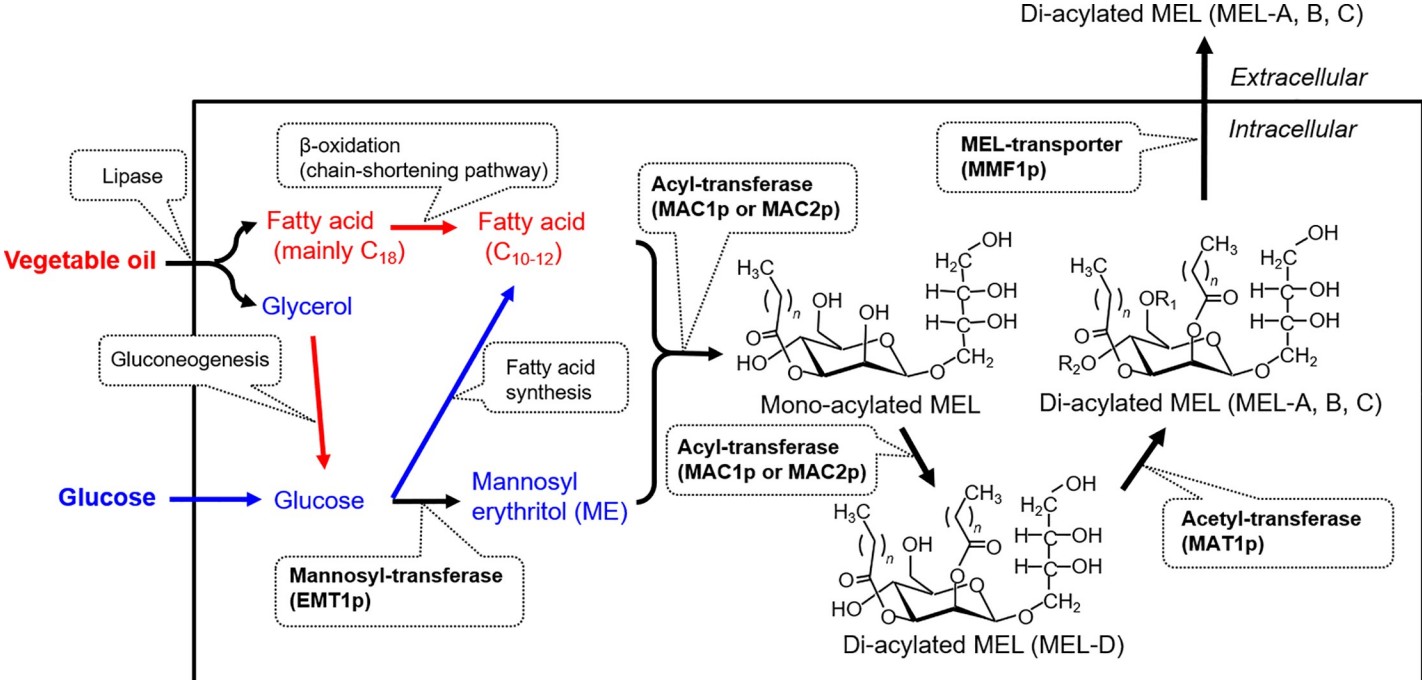

**Fig 1. Biosynthetic pathway for mannosylerythritol lipids.** Emt1p: mannose/erythritol transferase; Mac1p and Mac2p: acyl-transferases; Mat1p: acetyl-transferase; Mmf1p: predicted MEL transporter.

(UM521) was obtained from the Deutsche Sammlung von Mikroorganismen und Zellkulturen GmbH, Braunschweig, Germany. Stock cultures were cultivated for 3 days at 25˚C in YM plate medium containing 1% glucose, 0.5% peptone, 0.3% yeast extract, 0.3% malt extract, and 3.0% agar. The cultures were stored at 4˚C and renewed every 2 weeks.

## Culture conditions

Seed cultures were prepared by inoculating cells grown on slants into test tubes containing YM medium at 25˚C on a rotary shaker (250 rpm) for 2 days. Seed cultures (1 mL) were transferred to test tubes containing 20 mL of an experimental medium (0.03% $MgSO_4$, 0.03% $KH_2PO_4$, 0.1% yeast extract, pH 6.0) containing different carbon sources, and were then incubated as above for 5 days.

## Isolation and detection of glycolipids

After cultivation, the whole culture, including glycolipids, was extracted using an equal volume of ethyl acetate. The ethyl acetate extracts were analyzed using thin layer chromatography (TLC). TLC was performed using chloroform-methanol-7 N ammonium hydroxide (65:15:2, vol/vol/vol) as the solvent system. Although not a quantitative method, the MEL production was evaluated by TLC analysis in order to make a simple visual comparison of differences in MEL production depend on the producers and substrates. Visualization was performed by spraying 0.3% anthrone-sulfate reagents on the TLC plate and heating it at 90˚C for 5 min [4]. Purified MEL-A, MEL-B, and MEL-C, which were prepared from soybean oil by *P. antarctica* T-34, were used in the following experiments as standards.

## Estimation of cell growth

To estimate the effect of cell growth on different substrates, MELs and residual hydrophobic substrates were removed from culture broths by ethyl acetate. Then, the culture broth was washed twice with methanol to remove residual ethyl acetate. Cells were suspended in water of the same volume as the culture broth. The suspensions were centrifuged at $6,000 \times g$, and the pellets were weighed. All measurements were calculated from at least three independent experiments.

## Enzyme assay

The cells were harvested by centrifugation, washed, and suspended in 50 mM sodium acetate buffer (pH 5.0) containing 2 mM phenylmethylsulfonyl fluoride. The cells were disrupted in a 2 mL microtube with 0.5 mm glass beads using the BEADS CRUSHER μT-12 (TAITEC, Tokyo, Japan). The resulting homogenate was centrifuged at $20,000 \times g$ for 20 min to remove the glass beads and cell debris, and the supernatant was used as the crude enzyme solution.

Isocitrate dehydrogenase was measured in 96-well plates containing 0.2 mL of a reaction mixture consisting of 40 mM Tris-HCl (pH 7.0), 1 mM tri-sodium-isocitrate, 4 mM $MgCl_2$, and 0.5 mM $NAD^+$. Reactions were started by adding the crude enzyme solution (5 μg), and the absorbance at 340 nm was read every 1 min for up to 20 min using Synagy2 (BioTek Instruments, Winooski, VT, USA). A control lacking tri-sodium-isocitrate was also run. Enzymatic activity was determined by measuring the maximum rate of NADH production [17]. Protein concentrations were determined using a BCA protein assay kit (Pierce, Rockford, IL, USA) with bovine serum albumin as a standard. All measurements were calculated from at least three independent experiments.

## Statistical analysis of DNA microarray data

Two values of the expression level and induction of each gene by each experiment using a two-color platform were obtained from previous microarray data [16]. The Seris accession number in the Gene Expression Omnibus for the gene expression data was GSE47171. In the data, duplicated measurements were performed for each sample using the dye-swap method. Data were normalized by setting the baseline measurement per spot and per segment of the chip via intensity dependent (LOESS) normalization using the marray module of the R package [18] and Bioconductor [19]. Duplicated data files generated by the dye-swap experiment for an individual test were merged using the limma module. The logarithmic induction ratio (M-value) was calculated using the following equation:

$$M-value = \log_2(Cy5) - \log_2(Cy3) \tag{1}$$

and the average of the logarithmic signal intensities (A-value) was calculated using the following equation:

$$A-value = 0.5[\log_2(Cy5) + \log_2(Cy3)] \tag{2}$$

where Cy5 and Cy3 are normalized intensities for each transcript under the respective conditions using soybean oil or glucose as the sole carbon source [20]. The *P. antarctica* T-34 genes showing upregulated transcriptional expression in the presence of vegetable oil as compared to glucose were selected as those upregulated under oily conditions.

## Results

### *Pseudozyma antarctica* genes expressing under oily conditions

To demonstrate the transcriptomic regulatory characteristics of *P. antarctica* T-34 under oily conditions that result in the production of MELs, we focused on genes that were differentially regulated between *P. antarctica* T-34 and *U. maydis* UM521, using the DNA microarray data. Previously, the microarray data were derived from the transcriptomes of each strain cultured in medium containing 5% soybean oil or 10% glucose as the sole carbon source for 3 days. During this cultivation, larger amounts of MELs (30 g/L of MELs, mainly MEL-A) were produced from soybean oil by *P. antarctica* T-34, whereas *U. maydis* UM521 produced only small quantities of MELs on TLC analysis, and considerable soybean oil remained in the culture [16]. MEL production by *P. antarctica* T-34 from glucose (5 g/L of MELs, mainly MEL-A and mono-acylated type of MELs) was lower than that from soybean oil. Transcriptional expression analysis was previously performed using a two-color microarray platform, and the relative abundances of each transcript were compared between cells grown in the presence of soybean oil or glucose as the sole carbon source. Values, the logarithmic induction ratio (M-value), average of the logarithmic signal intensities (A-value) and probability values (p-value), were obtained after processing the raw DNA microarray data [16].

As described previously [15], in *P. antarctica* T-34, the MEL biosynthesis gene cluster was identified (**Fig 2A**) as being expressed under both conditions, although the production level was higher in the presence of excess vegetable oil. The gene expression intensity (A-value) and induction ratio (M-value) of the five genes were significantly higher in *P. antarctica* T-34 than in *U. maydis* UM521 [16]. In the present study, we further analyzed the data and visualized the distribution of gene expression and the induction ratio by vegetable oil as a two-dimensional histogram, which revealed that *U. maydis* UM521 genes were widely scattered compared with those of *P. antarctica* T-34 (**Fig 2B and 2C**). To estimate the transcriptional regulatory characteristics of both strains under oily conditions, genes showing expression patterns similar to the MEL biosynthesis genes shown in **Fig 1B and 1C** were selected and analyzed. The 264 genes with an A-value greater than 14 in *P. antarctica* and an M-value less than 0 for the ortholog in *U. maydis* are listed in **Table 1 and S1 Table**. As expected, the four genes for MEL biosynthesis, *PaEMT1* (19c00001), *PaMAC1* (19d00003), *PaMMF1* (19d00004), and *PaMAT1* (19c00002), were included in the list.

Based on the Eukaryotic Orthologous Groups classification, 179 of the 264 genes were assigned to each category. 93 of the genes were assigned to the metabolism category, corresponding to 35.2% of the 264 genes, whereas metabolism related genes comprised only 17.7% of the total coding sequence (CDS) (1,157 genes). However, the proportions of genes categorized as cellular processes (15.9%), information storage and processing (6.1%), and poorly characterized (10.6%) were lower than those of the total CDS (**Table 1**).

Within the metabolism category, the proportions of genes classified as amino acid, carbohydrate, inorganic ions, and secondary metabolite metabolism, as well as energy production and conversion, were higher among the listed 264 genes than among the total CDS, whereas the proportions involved in lipid transport and metabolism were almost identical between the listed genes (3.8%) and total CDS (3.3%). The listed genes responsible for metabolism, with the exception of lipid metabolism, were therefore expressed along with MEL biosynthesis genes under oily conditions, suggesting that *P. antarctica* T-34 efficiently generated energy from vegetable oil via respiratory metabolism. In addition, the genes related to glycolysis and the TCA cycle, which are part of the central metabolic pathway, may relate to MEL production, because the transcriptional regulatory characteristics are similar to MEL biosynthesis genes.

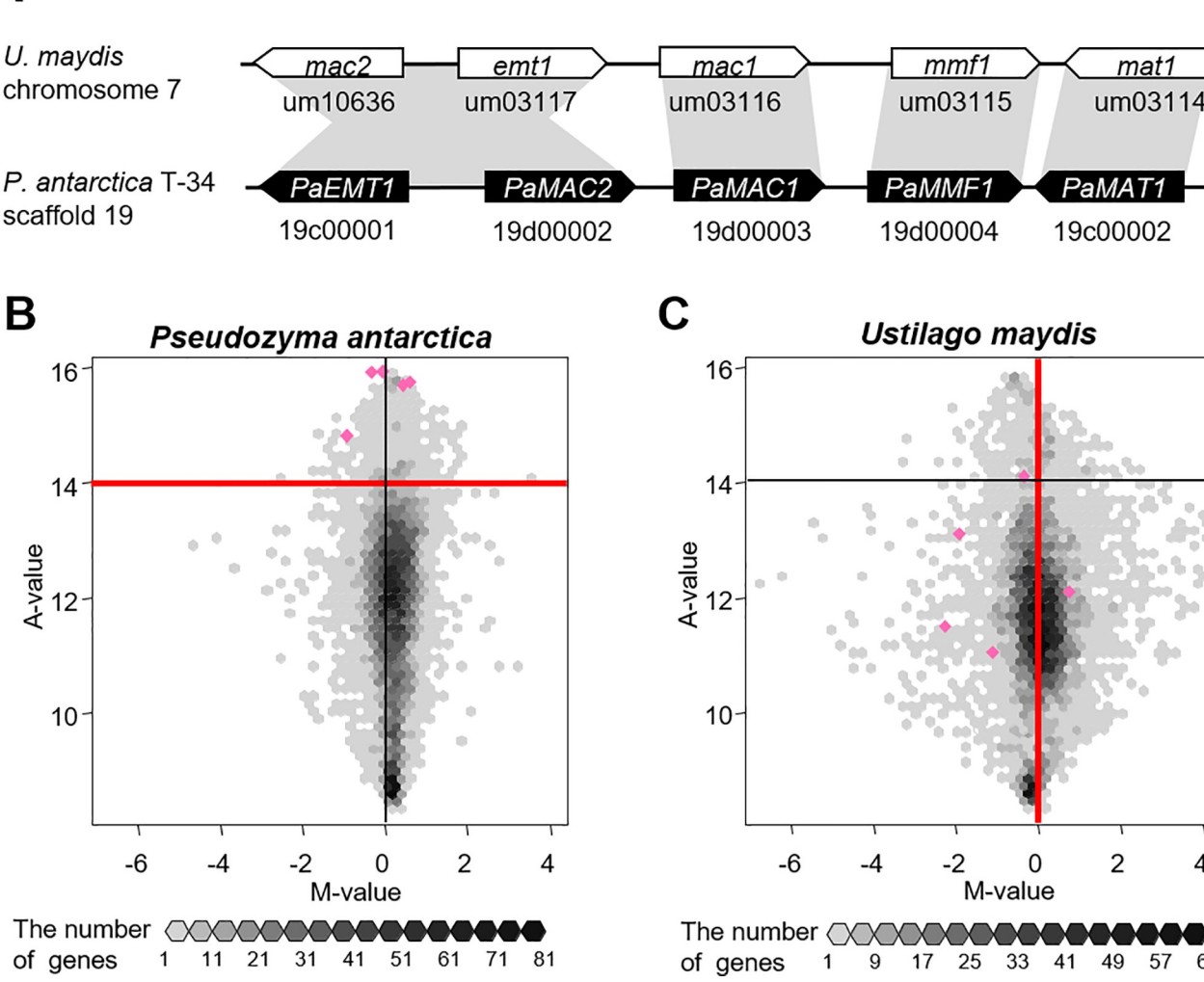

**Fig 2. Gene clusters of mannosylerythritol lipid (MEL) biosynthesis and gene expression profiles.** The MEL biosynthesis gene cluster in *U. maydis* UM521 and *P. antarctica* T-34 (A). Two-dimensional histogram showing the gene expression distribution (A-value, ordinate) and gene transcription induction ratio (M-value, abscissa) in the presence of vegetable oil and glucose in *P. antarctica* T-34 (B) and *U. maydis* UM521 (C). The number of genes is represented by the gray shading of each hexagonal cell. This hexplot was drawn using the hexbin module in R [21]. The genes responsible for MEL biosynthesis are overlaid on the hexplot in pink.

## Metabolic genes involved in MEL production

As detailed above, the genes related to metabolism, e.g., beta-oxidation, fatty acid synthesis, glycolysis/gluconeogenesis, and TCA cycle, were predominantly expressed during MEL production. The genes involved in beta-oxidation and fatty acid synthesis were extracted from the Saccharomyces Genome Database, and the orthologs were selected using a homology search (S2 and S3 **Tables**). The genes involved in peroxisomal and mitochondrial beta-oxidation were induced in both *P. antarctica* T-34 and *U. maydis* UM521, whereas the induction ratios of the genes for fatty acid synthesis were suppressed in both strains (**Fig 3**). These results suggest that both strains degraded vegetable oil via the beta-oxidation pathway, and that fatty acid synthesis was required in the presence of glucose because fatty acid must be synthesized from glucose to supply cellular demands and for MEL biosynthesis. In addition, M-value average of MEL biosynthesis genes were −0.499 and −1.38 in *P. antarctica* T-34 and *U. maydis* UM521,

**Table 1. KOG classification of 264 genes extracted by guilt-by-association.**

| Category | Description | The number of genes | Ratio (%) | The number of genes "picked" | Ratio (%) |
|---|---|---|---|---|---|
| Metabolism | | | | | |
| C | Energy production and conversion | 206 | 3.1 | 20 | 7.6 |
| E | Amino acid transport and metabolism | 204 | 3.1 | 18 | 6.8 |
| F | Nucleotide transport and metabolism | 63 | 1.0 | 1 | 0.4 |
| G | Carbohydrate transport and metabolism | 172 | 2.6 | 19 | 7.2 |
| H | Coenzyme transport and metabolism | 78 | 1.2 | 4 | 1.5 |
| I | Lipid transport and metabolism | 219 | 3.3 | 10 | 3.8 |
| P | Inorganic ion transport and metabolism | 91 | 1.4 | 9 | 3.4 |
| Q | Secondary metabolites biosynthesis, transport and catabolism | 124 | 1.9 | 12 | 4.5 |
| | Subtotal | 1157 | 17.7 | 93 | 35.2 |
| Cellular processes | | | | | |
| D | Cell cycle control, cell division, chromosome partitioning | 139 | 2.1 | 0 | 0.0 |
| M | Cell wall/membrane/envelope biogenesis | 48 | 0.7 | 3 | 1.1 |
| N | Cell motility | 3 | 0.0 | 0 | 0.0 |
| O | Posttranslational modification, protein turnover, chaperones | 373 | 5.7 | 21 | 8.0 |
| T | Signal transduction mechanisms | 29 | 4.4 | 2 | 0.8 |
| U | Intracellular trafficking, secretion, and vesicular transport | 256 | 3.9 | 9 | 3.4 |
| V | Defense mechanisms | 31 | 0.5 | 2 | 0.8 |
| W | Extracellular structure | 4 | 0.1 | 0 | 0.0 |
| Y | Nuclear structure | 24 | 0.4 | 0 | 0.0 |
| Z | Cytoskeleton | 95 | 1.4 | 5 | 1.9 |
| | Subtotal | 1,264 | 19.3 | 42 | 15.9 |
| Information storage and processing | | | | | |
| A | RNA processing and modification | 203 | 3.1 | 2 | 0.8 |
| B | Chromatin structure and dynamics | 78 | 1.2 | 1 | 0.4 |
| J | Translation, ribosomal structure and biogenesis | 287 | 4.4 | 11 | 4.2 |
| K | Transcription | 216 | 3.3 | 0 | 0.0 |
| L | Replication, recombination and repair | 164 | 2.5 | 2 | 0.8 |
| | Subtotal | 948 | 14.5 | 16 | 6.1 |
| Poorly characterized | | | | | |
| R | General function prediction only | 588 | 9.0 | 23 | 8.7 |
| S | Function unknown | 244 | 3.7 | 5 | 19 |
| | Subtotal | 832 | 12.7 | 28 | 10.6 |
| The genes categorized with KOG classification | | 4,201 | 64.1 | 179 | 67.8 |
| Total CDS | | 6,555 | 100.0 | 264 | 100.0 |

respectively. This result indicates that MEL biosynthesis pathway in *U. maydis* UM521 is suppressed compared to that in *P. antarctica* T-34.

In addition, a gene encoding an extracellular lipase (12c00005), which is required for effective oil degradation [22], was expressed (A-value: 12.0) and strongly induced (M-value: 1.70) in *P. antarctica* T-34. However, the orthologous gene (um03410.1, accession #: Q4P903) was slightly suppressed in *U. maydis* UM521 (M-value: −0.28), and the expression intensity (A-value: 9.98) was significantly lower. This gene was likely a major factor in the more effective utilization of vegetable oil in *P. antarctica* T-34 than in *U. maydis* UM521.

The expression profiles of the genes responsible for glycolysis and the TCA cycle differed between *P. antarctica* T-34 and *U. maydis* UM521 (**Fig 3**). Notably, the average logarithmic induction ratio (M-value) of most genes in the central metabolic pathway, such as those

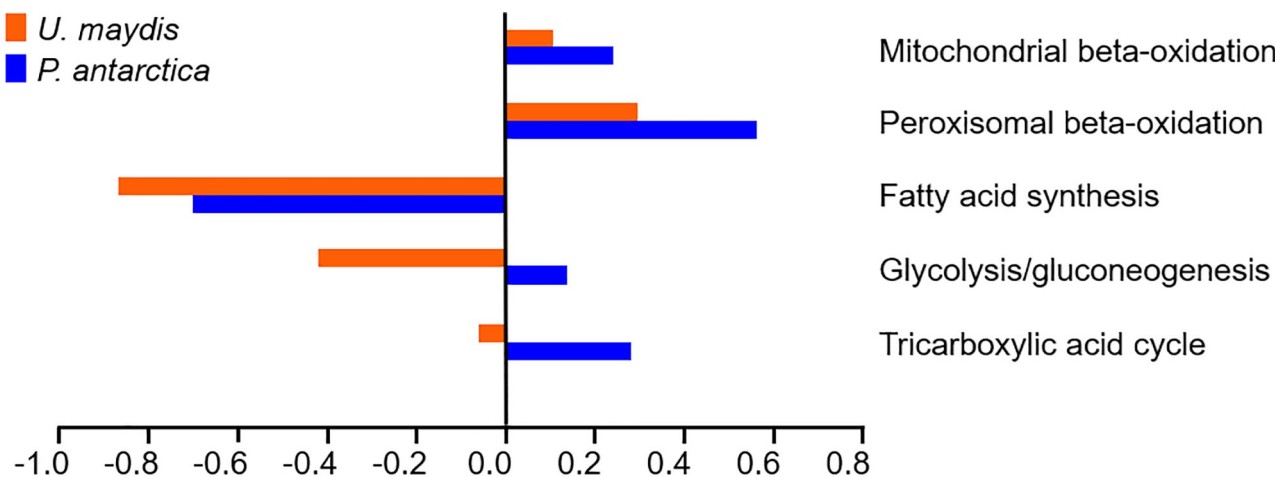

Fig 3. Induction of genes responsible for oil degradation and conversion and primary metabolism. The M-value averages of the genes responsible for MEL biosynthesis, mitochondrial beta-oxidation, peroxisomal beta-oxidation, fatty acid synthesis, glycolysis, tricarboxylic acid cycle, pentose phosphate pathway, and malate/pyruvate cycle are shown in orange (*U. maydis* UM521) and blue (*P. antarctica* T-34).

involved in glycolysis and the TCA cycle, in *P. antarctica* T-34 was positive, whereas that in *U. maydis* UM521 was negative. We thus further analyzed the detailed expression levels of glycolysis and TCA cycle genes.

## Glycolysis and TCA cycle genes

As shown in **Fig 4**, the M-values of all genes related to the TCA cycle in *P. antarctica* T-34 were positive, indicating that these genes were induced in the presence of vegetable oil, whereas the M-values in *U. maydis* UM521 close to zero. The genes encoding isocitrate lyase (22c00252, M-value: 0.78; A-value: 13.2) and malate synthase (14d00059, M-value: 1.3; A-value: 15.0) as the glyoxylate shunt, and malic enzyme (7c00096, M-value: 0.43; A-value: 14.8) and phosphoenolpyruvate carboxykinase (2d00021, M-value: 1.1; A-value: 15.3) as the anaplerotic reaction, were highly expressed under oily conditions in *P. antarctica* T-34. These results indicate that the glyoxylate shunt and the anaplerotic reaction assisting gluconeogenesis for sugar synthesis in *P. antarctica* T-34 were active under oily conditions. However, the M-values of these genes were also positive in *U. maydis* UM521, whereas the citrate synthase catalyzing primary reactions of the TCA cycle were strongly repressed (M-value: −0.88). These results suggest that the glyoxylate shunt and the anaplerotic reaction were inactive in metabolic flow, although each gene was expressed under oily conditions in *U. maydis* UM521. Moreover, most genes related to glycolysis were highly expressed even under oily conditions in *P. antarctica* T-34, whereas those in *U. maydis* UM521 were repressed.

## Activities of the TCA cycle enzymes in *P. antarctica* and *U. maydis*

Among TCA cycle gene expression, the M-value of the gene encoding isocitrate dehydrogenase, which is a rate-limiting enzyme of the cycle, in *P. antarctica* T-34 was positive, whereas that in *U. maydis* UM521 was negative. We thus measured the enzyme activity to further estimate the function of the TCA cycle based on the enzymatic activities in *P. antarctica* T-34 and *U. maydis* UM521 under oily conditions. Both strains were cultured in medium containing 5%

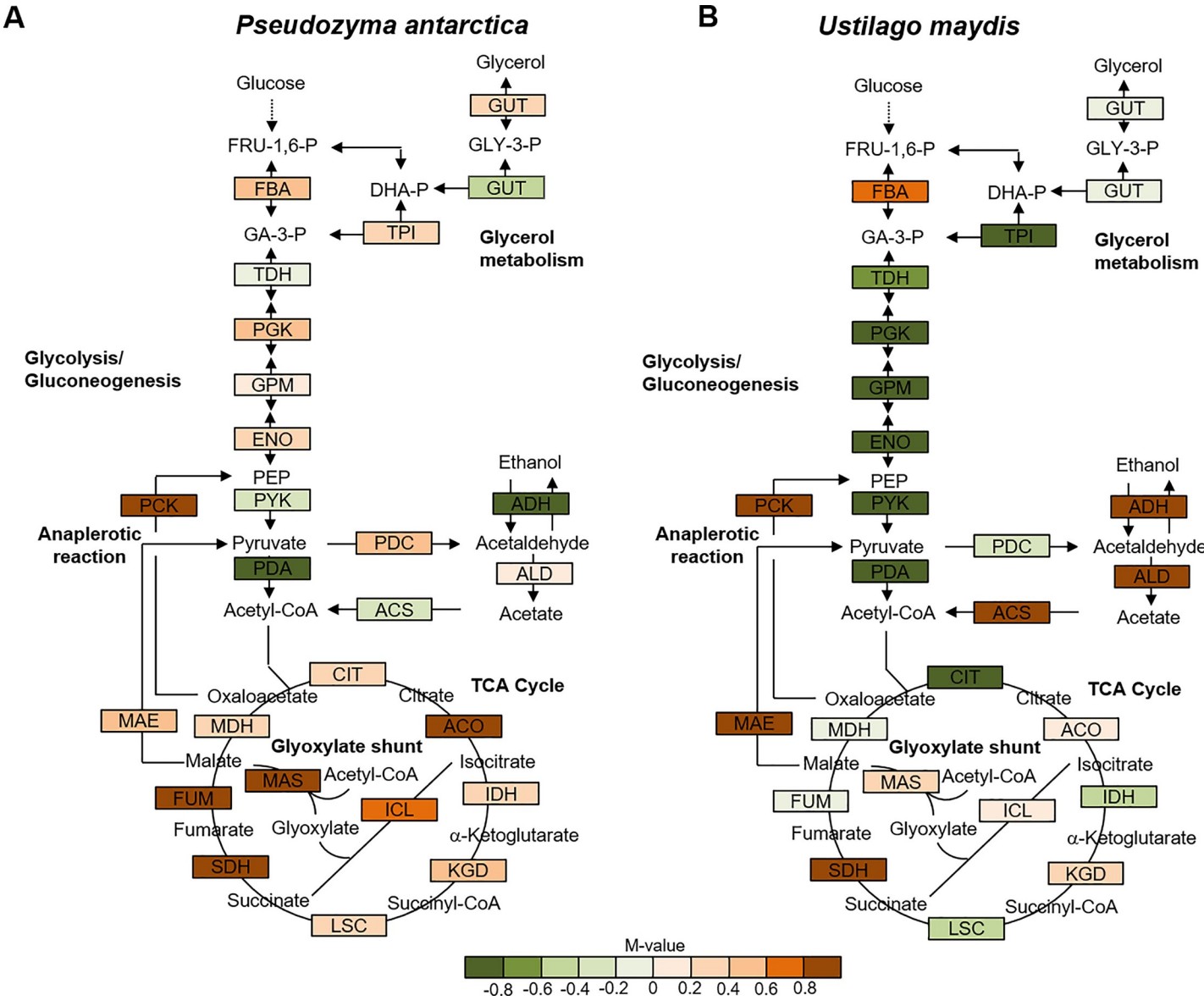

**Fig 4. M-values in the central metabolic pathway of *P. antarctica* T-34 and *U. maydis* UM521.** Comparison of the expression profiles of genes encoding enzymes that participate in glycolysis and the tricarboxylic acid cycle under oily conditions in *P. antarctica* T-34 (A) and *U. maydis* UM521 (B). To facilitate the comparison of the two microorganisms, portions of the metabolic pathways of *Saccharomyces cerevisiae* are presented, and the behaviors of genes encoding the enzymes that catalyze each step were estimated. GUT: glycerol-3-phosphate dehydrogenase; TPI: triose phosphate isomerase; FBA: fructose 1,6-bisphosphate aldolase; TDH: glyceraldehyde-3-phosphate dehydrogenase; PGK: 3-phosphoglycerate kinase; GPM: phosphoglycerate mutase; ENO: enolase; PYK: pyruvate kinase; PDA: pyruvate dehydrogenase; PDC: pyruvate decarboxylase; ACS: acetyl-coA synthetase; ADH: alcohol dehydrogenase; ALD: aldehyde dehydrogenase; PCK: phosphoenolpyruvate carboxykinase; MAE: malic enzyme; CIT: citrate synthase; ACO: aconitase; IDH: isocitrate dehydrogenase; KGD: α-ketoglutarate dehydrogenase; LSC: succinyl-CoA ligase; SDH: succinate dehydrogenase; FUM: fumarate hydratase; MDH: malate dehydrogenase; ICL: isocitrate lyase; MAS: malate synthase. Red and green boxes represent those genes whose expression levels were increased and decreased, respectively, under oily conditions.

soybean oil or 10% glucose as the sole carbon source for 5 days. Glycolipids were then extracted using an equal amount of ethyl acetate, and MEL production was confirmed by TLC analysis using the anthrone staining method. *P. antarctica* T-34 produced larger amounts of MELs, mainly MEL-A, with soybean oil but lower quantities from glucose. During cultivation, almost all of the soybean oil was consumed by *P. antarctica* T-34 (**Fig 5A**). In contrast, *U. maydis* UM521 produced lower amounts of MELs, and larger amounts of soybean oil remained in

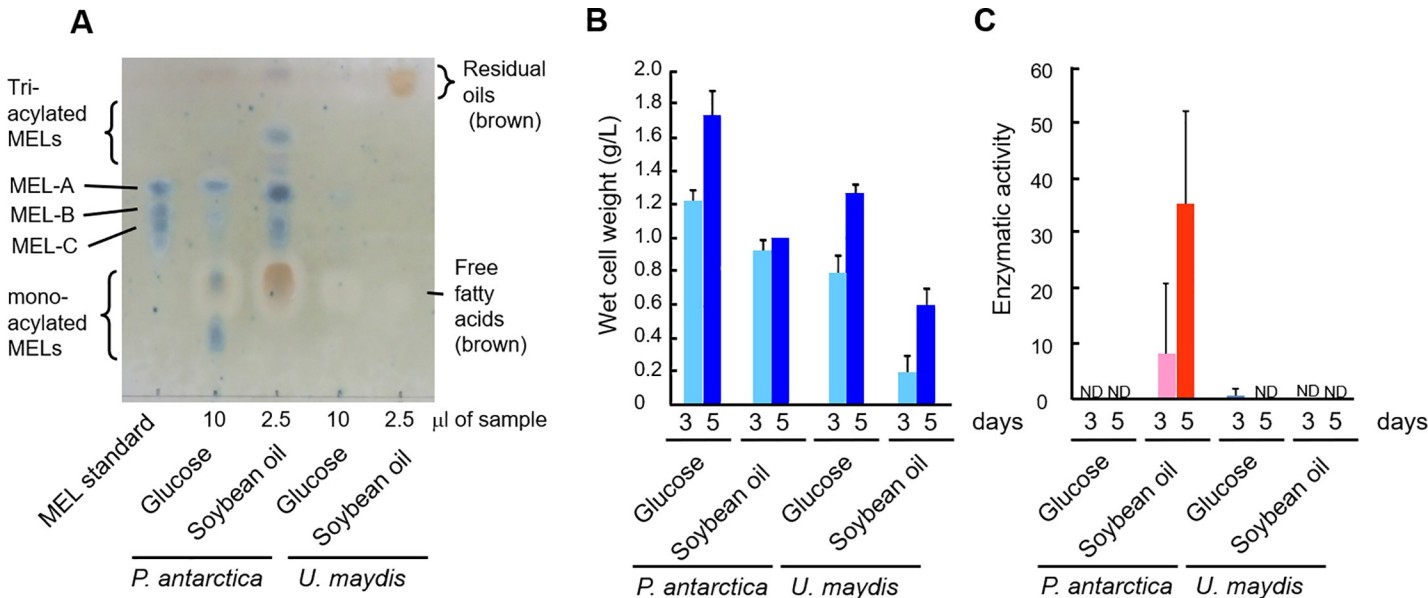

**Fig 5. Comparison of mannosylerythritol lipid (MEL) production, cell growth, and isocitrate dehydrogenase (NAD$^+$) activity of *P. antarctica* T-34 and *U. maydis* UM521 in the presence of glucose or soybean oil as the sole carbon source.** (A) MELs were extracted from the culture after 3 and 5 days using an equal amount of ethyl acetate, and the organic solvent fractions (with glucose: 10 µL, with soybean oil: 2.5 µL) were spotted on a TLC plate. A typical sample of three independent experiments was shown in each lane. Spots were visualized using the anthrone reagent. Purified MEL-A, MEL-B, and MEL-C were used as standards. (B) The weight of wet cells was measured to estimate growth. (C) The activity of isocitrate dehydrogenase (NAD$^+$) was measured as described in the Materials and methods section. Error bars show standard deviations.

the medium. *U. maydis* UM521 grew slowly under the oily conditions compared with *P. antarctica* T-34 (**Fig 5B**). These results of MEL production characteristics of both of strains are identical to previous study [16].

Isocitrate dehydrogenase activity was induced in the T-34 strain of *P. antarctica* under oily conditions, whereas no activity was detected in the presence of glucose (**Fig 5C**). In *U. maydis* UM521, the activity was low in the presence of glucose, and no activity was noted under oily conditions (**Fig 4**).

## Discussion

Transcriptomic analyses of associated genes and pathways provided new insight into the capacity of *P. antarctica* T-34 to produce functional bio-based materials from oil biomass. Surprisingly, the genes responsible for central metabolic pathways such as glycolysis, the TCA cycle, the glyoxylate shunt, and the anaplerotic reaction were highly expressed in *P. antarctica* T-34, regardless of whether the carbon source was glucose or soybean oil, whereas these genes were not induced in *U. maydis* UM521, which could be the reason for poor growth and lower MEL production under oily conditions. The characteristics of *P. antarctica* T-34 that enabled it to efficiently metabolize vegetable oil via the central metabolic pathway facilitated the production of large amounts of extracellular MELs. In the future, *P. antarctica* will be increasingly used as a platform for new biomaterial production processes, such as producing functional lipids from oily biomass.

Previously, we reported that the genome sequence of *P. antarctica* T-34 was closely related to that of *U. maydis* UM521, based on the results of synteny analysis [16]. However, gene set enrichment analysis of the transcriptome revealed that gene expression differed markedly between *P. antarctica* T-34 and *U. maydis* UM521 in the presence of glucose. The present

analysis focused on the expression of genes required for cell growth under oily conditions, which prompted us to further investigate the prerequisites for extracellular production of MELs utilizing large quantities of vegetable oil.

In *P. antarctica* T-34, fatty acids derived from vegetable oil by enzymatic degradation were processed via chain-shortening pathways such as beta-oxidation, and the intermediates were directly transferred into mannosylerythritol, resulting in production of MELs [23]. However, *U. maydis* UM521 did not metabolize vegetable oil, resulting in poor growth and low MEL productivity (**Fig 5**) identical to our previous report [16]. This phenomenon was supported by the present transcriptomic analysis, showing that the expression levels of most genes of central metabolic pathways were reduced under oily conditions in *U. maydis* UM521.

However, the genes encoding enzymes in the TCA cycle (citrate synthase, aconitase, succinate dehydrogenase, fumarase, and malate dehydrogenase), glyoxylate shunt (isocitrate lyase and malate synthase), and anaplerotic reaction (malic enzyme and phosphoenolpyruvate carboxykinase) were highly expressed in the presence of vegetable oil in *P. antarctica* T-34 (**Fig 4**). A similar process has been reported in *Escherichia coli* when cultured with acetate or fatty acid as a sole carbon source [24, 25]. In addition, $^{13}$C-metabolic flux analysis revealed that the glyoxylate shunt was activated in *Yarrowia lipolytica* for growth and lipids production when cultured with acetic acid, which is kind of volatile fatty acid, as sole carbon source [26]. Importantly, the glyoxylate shunt and anaplerotic reaction contribute to carbon assimilation from acetyl-CoA and effective carbon delivery into the gluconeogenesis pathway, respectively. Additionally, enhancement of these primary metabolic activities may lead to improved sugar synthesis for cell growth and MEL production from vegetable oil. Therefore, although the direction of the carbon flow was unclear in the results of transcriptomic analysis, most reactions in the glycolytic process are reversible, and it is probable that the enhanced expression of glycolysis genes reflected the activation of gluconeogenesis under oily conditions. Further metabolomics studies are needed to reveal the carbon flow in the primary metabolic pathways under oily conditions. Although *P. antarctica* T-34 and *U. maydis* UM521 are closely related species, characteristics of growth and MEL production, including the activities of central metabolic pathways, differ between them. We hypothesize that these differences originated in specific growth environments and lifestyles. The genus *Pseudozyma* is often isolated from plant surfaces in nature [4, 27, 28], and utilizes cuticles composed of water insoluble fatty acid esters covering leaves with their lipases and esterases [29]. Genus *Pseudozyma* may therefore utilize the secretion of biosurfactants such as MELs to uptake emulsified cuticles as a carbon source. However, *U. maydis* UM521 is a maize pathogen, and can utilize starch-derived sugar, which is abundant in the growing environment. Our transcriptomic analysis suggested that sufficient primary metabolic activity was required for growth and MEL production under oily conditions, in addition to expression of the genes responsible for MEL biosynthesis and oil degradation in *P. antarctica*. These results indicate that *U. maydis*, unlike *P. antarctica*, adapted to utilize sugars as the main carbon source.

Moreover, 111 of the 264 genes in **S1 Table** had no clear assignment, and were listed as "function unknown," "general function predicted only", or "no description." We further categorized these genes based on the Pfam database. The gene functions of 88 of the genes were predicted, but the remaining 23 genes were uncharacterized. Fifty-three of the *P. antarctica* T-34 genes nonorthologous to *U. maydis* UM521 are also listed in S4 **Table**. Genes contributing to oil utilization may be hidden among them. Further studies related to these genes will help identify fundamental differences in the metabolic systems between nonpathogenic *P. antarctica* T-34 and plant-pathogenic *U. maydis* UM521.

Various vegetable oils have been used as main carbon sources for MEL production, due to high productivities, e.g., *P. aphidis* produced 165 g/L of MELs (the main product is MEL-A)

from soybean oil and glucose, and *P. regulosa* produced 142 g/L of MELs (the main product is MEL-A) from soybean oil and erythritol, *P. tsukubaensis* produced 73.1 g/L of MELs (the main product is MEL-B) from soybean oil. While *U. maydis* also produced MELs from sunflower oil, the productivity was lower than that of the genus *Pseudozyma* due to the products were a mixture of MELs and CLs. On the other hand, soluble carbon sources such as glycerol, glucose, and sucrose were examined for MEL production, e.g., *P. antarctica* produced mono-acylated type of MEL as the main product from glucose [10], and *U. maydis* produced 32.1 g/L of a glycolipid mixture of MELs and CLs from glycerol [30]. Therefore, vegetable oils are sufficient carbon sources for the MEL production. Recently, we accomplished with improvement of MEL production from olive oil by increase of the expression of a gene encoding lipase in *P. tsukubaensis* [22].

The present transcriptomic and biochemical analyses focused on gene expression in *P. antarctica* T-34, a highly oil-assimilating and glycolipid-producing yeast, in the presence of vegetable oil. Further genetic study will facilitate not only the development of useful industrial strains as MEL producer by genetic modification, but also improve our understanding of phytopathological mechanisms and environmental adaptation of these basidiomycetous genera.

## Supporting information

**S1 Table. The list of 264 genes extracted by guilt-by-association.**
(XLSX)

**S2 Table. The list of the genes related to lipogenesis pathway in *P. antarctica* T-34.**
(XLSX)

**S3 Table. The list of the genes related to lipogenesis pathway in *U. maydis* UM521.**
(XLSX)

**S4 Table. The list of the non-ortholog genes expressing and inducing under oily conditions in *P. antarctica* T-34.**
(XLSX)

## Acknowledgments

We thank all the members of the Bio-chemical Group at AIST.

## Author Contributions

**Conceptualization:** Hideaki Koike, Tomotake Morita.

**Data curation:** Keisuke Wada, Hideaki Koike.

**Formal analysis:** Hideaki Koike.

**Investigation:** Tomotake Morita.

**Writing – original draft:** Keisuke Wada.

**Writing – review & editing:** Tatsuya Fujii, Tomotake Morita.

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
