## [Decision Letter · Decision Letter 0]

6 Sep 2019

PONE-D-19-22493

A novel aspect of the production of mannosylerythritol lipids in Pseudozyma antarctica T-34 based on gene expression of central metabolic pathways

PLOS ONE

Dear Dr Morita,

Thank you for submitting your manuscript to PLOS ONE. After careful consideration by four experts in the field, we feel that it has merit but does not fully meet PLOS ONE’s publication criteria as it currently stands. Therefore, we invite you to submit a revised version of the manuscript that addresses the various rather minor points raised during the review process.

We would appreciate receiving your revised manuscript by the end of september. To enhance the reproducibility of your results, we recommend that if applicable you deposit your laboratory protocols in protocols.io, where a protocol can be assigned its own identifier (DOI) such that it can be cited independently in the future. For instructions see: http://journals.plos.org/plosone/s/submission-guidelines#loc-laboratory-protocols

We look forward to receiving your revised manuscript.

Kind regards,

Marie-Joelle Virolle, PhD

Academic Editor

PLOS ONE

Journal Requirements:

- Tomotake Morita, Hideaki Koike, Hiroko Hagiwara, Emi Ito, Masayuki Machida, Shun Sato, Hiroshi Habe, Dai Kitamoto. "Genome and Transcriptome Analysis of the Basidiomycetous Yeast Pseudozyma antarctica Producing Extracellular Glycolipids, Mannosylerythritol Lipids", PLoS ONE, 2014

- https://www.ncbi.nlm.nih.gov/pmc/articles/PMC3622993/

In your revision ensure you cite all your sources (including your own works), and quote or rephrase any duplicated text outside the methods section. Further consideration is dependent on these concerns being addressed.

Reviewers' comments:

Reviewer's Responses to Questions

**Comments to the Author**

1. Is the manuscript technically sound, and do the data support the conclusions?

Reviewer #1: Partly

Reviewer #2: Yes

Reviewer #3: Yes

Reviewer #4: Yes

2. Has the statistical analysis been performed appropriately and rigorously? 

Reviewer #1: No

Reviewer #2: Yes

Reviewer #3: No

Reviewer #4: Yes

3. Have the authors made all data underlying the findings in their manuscript fully available?

Reviewer #1: Yes

Reviewer #2: Yes

Reviewer #3: Yes

Reviewer #4: Yes

4. Is the manuscript presented in an intelligible fashion and written in standard English?

Reviewer #1: Yes

Reviewer #2: Yes

Reviewer #3: No

Reviewer #4: Yes

5. Review Comments to the Author

Reviewer #1: Linguistic comments

• Line 65: “large amounts of vegetable oil”. Change it to “vegetable oil”.

• Line 153: “a large quantities of MELs.” Change it to “MELs” or define large quantities of MELs

• Line 197: Remove “surprisingly”

• Line 251: “were almost negative”. Change it to “close to zero”.

• Line 289: “enzymes”. Change it to “gene expressions”.

• Line 314: “strongly induced”. Change it to “induced”.

• Line 338 & 352: “large quantities”; Change it to “larger quantities”.

• General: When adding production numbers, they should be mentioned in volumetric productivities for improved comparison in literature.

Research comments

• There is no mentioning of any statistical significance of the genes or the selection procedure of the genes illustrated in figure 2 B&C. This part needs extended clarification.

• Line 177-181: What is the scientific background of this selection procedure for “similar genes expressions to the MEL biosynthesis genes”? Cut-offs of A-values > 14 and M < 0 for U. Maydis orthologs seem to be defined arbitrary. One of the MEL production genes has even a M>0. Revise the selection procedure or mention the scientific/statistical incentives for the cut-offs.

• There is no mention of exact replicate numbers for all the experiments performed.

• Line 310: Why was wet cell weight used and dry cell weight. Change in wet cell weight can be caused by changes in substrate concentrations or evaporation.

Conclusion comments

• Line 170: Can you compare A-values between two different organisms?

• Line 211-212: State with literature or experimental evidence why TCA and glycolysis are related to production of MELs.

• Line 265-267: It’s only a slight indication. There are more levels of regulation (post-transcriptional, translational, post-translational,…). This statement needs either metabolomic data or needs to be removed.

• Line 326: change “resulting in” to “which could be the reason for”.

• Discussion: Add comparisons with literature concerning the central metabolism of lipophilic fungi.

• Discussion: Add comparisons with literature concerning the carbon sources used for MEL production and their respective production numbers.

Figure comments

• Figure 1: Figure description doesn’t match the figure. Discrepant abbreviations and full names of enzymes.

• Figure 1: rarely does vegetable oil solely consist of C18 fatty acids. It should be made clear that the fed vegetable oil predominantly consists of C18 fatty acids.

• Figure 2 B&C: Density is not clear. Use a continuous colour scale.

• Figure 2: “The genes responsible for MEL biosynthesis are overlaid on the hexplot”. Mention they are coloured in pink.

• Figure 5 B: Can it be that the higher production is cause by higher biomass? Calculate specific productivities (g MEL/ (g Biomass*h).

• Figure 5 B & C: Define the error flags. Are they standard deviations, confidence intervals,…

Extra information/reference necessary

• Line 58-60: “Pseudozyma antarctica T-34 produces large amounts of MELs when grown in culture containing vegetable oil as the carbon source, and the production yield reaches 140 g/L using n-alkanes as the carbon source”. Define “large amounts” and state with relevant research.

• Line 63: Define “lower amounts” and state with relevant research

• Line 75: Define “more effective”.

• Line 124 & 127: “Tri-isocitrate”. Do you mean tri-sodium-isocitrate?

• Line 137: URL is redundant information

• Line 144: Add the type of normalization that was used. In case of normalization with certain household genes, mention the genes used.

• Line 158: “(more than 30 g/L)”. Mention the exact amount that was measured

• Line 159: amount of MELs produced by U. maydis should be defined

• Line 161: MEL production should be mentioned.

• Line 167: “As described previously”. Add reference..

• Line 344: Quantify “large quantities” of MELs.

Reviewer #2: In the manuscript entitled „A novel aspect of the production of mannosylerythreitol lipids in Pseudozyma antacrtica T-34 based on gene expression of central metabolic pathways“ the authors Wada et al. have analysed micro array data of two fungi (P. antarctica T-34 and Ustilago maydis 521) and two growth conditions (glucose and soybean oil).

P. antarctica T-34 has produced under soy bean oil much more MELs than under glucose whereas in U. maydis the MEL production under soybean oil growth conditions is heavily repressed.

The authors have used the KOG nomenclature to categorize all genes and have identified genes of the primary metabolism to be upregulated in soybean oil conditions in P. antarctica. In contrast, most of these genes are down-regulated in U. maydis.

Finally, the authors demonstrate that under soybean oil conditions isocitrate dehydrogenase expression is increased and this coincides with increased activity.

The current work is interesting and will allow further manipulation of P. antarctica T-34 to increase MEL biosynthesis.

But there are some points necessary to improve the manuscript, the tables and the figures:

The title starts with “A novel aspect..” Please specify or change the title.

Lane 20: industrial lipases, mannosyl…

Lane 85: please complete the sentence: “…adapted to aerobically…”

Lane 110: please specify the concentration of the NH4OH-solution used as solvent system

Lane 150: The title “Pseudozyma antarctica genes are expressed…” is confusing. What kind of genes do you mean? Not all genes are expressed with high intensity, are they?

Table 1: I guess that you have ordered the table according to the Alphabet of the description. I would prefer to have then the description in the first column (not in the second). For the metabolism section “Energy production…” should be reordered. The row “Total CDS” should be changed with the row “The genes categorized…”. You have mentioned in the entire table only the KOG classified genes (93+42+16+28=179)

Lane 241: remove “Consequently,”

Lane 258 spelling: Antarctica

Lane 297: “large amounts of MELs”: can you specify, which variants of MELs are produced?

Lane 311: …isocitrate dehydrogenase (NAD+)…” In the figure you use MaxV. Please clearify.

Lane 317: “…no activity was noted under oily conditions.

Figure 1: According to the legend (Lanes 54-56) the proteins are assigned differently (Emt1p not EMT1 and so on). The proteins involved in the step from mono-acylation to di-acylation are the acyl-transferases MAC1 and MAC2 (Mac1p/Mac2p). In your scheme acylation of C3 (by Mac2p) precedes the acylation of C2 (by Mac1p). Please comment on that in the introduction.

Figure 3: “vegetable oil over glucose”

Figure 4: Great work!!

Figure 5:

A: Please add MEL-A, -B, -C; the curly bracket for MELs is huge. Are you sure, that the slow migrating bands are MELs?

Why you mention “Olive oil” and not “Soybean oil”?

B: again “Olive oil”

C: again “Olive oil”. I propose to use different colors as in 5B, because you performed a different experiment.

Lane 339-344: I think you can remove the sentence: “Hence, ….quantities of MELs”. To my opinion it repeats the sentence before.

Lane 383: “..will help to identify…”

Supplemental tables:

As for table 1: The sorting criteria must be in the first column.

S1: Start with KOG, KOG number, Pa gene…

S2 and S3: please use either upper- or lowercase names for the enzyme description and resort according to Alphabet, then EC-number…

Legend: …*3 BBH means bi-directional best hit. Plural means the homology…” Please change all “pulural” to plural in column Type of S2

S4: please sort the gene numbers: 1, 2, 3, 4, 5, 6, 7, 8, 9, 10, 11, and so on

Start with: 1c00036, 1d00040, 2c00011 and so on

The description column uses cryptic abbreviations e.g. row 9: Aldedh ? Please use for all domain abbreviations the full form.

Reviewer #3: The manuscript "A novel aspect of the production of mannosylerythritol lipids in Pseudozyma antarctica

T-34 based on gene expression of central metabolic pathways" gives new insights on the genetic and metabolic regulations controlling mannosylerythritol lipids (MEL) biosynthesis in two yeasts.

The manuscript is providing a new piece o knowledge ans is sound for publications, yet after considering the following comments:

1- Lines 344-345: The authors mention that the Ustilago maydis had poor growth when using vegetable oils as carbon source, which is not the case with Pseudozyma antarctica. I strongly suggest the absence of lipases and esterases in U. maydis which explain their poor growth and low MEL production. Therefore, I suggest the authors screen the genome of U. maydis for lipases and esterases and if absent, to use this as the explanation of the poor growth and low MEL production. Also, does U. maydis grow on alkanes or fatty acids or glycerol separately?

2- Line 67-68: Although authors are referring to reference 15, yet it would be easier for the reader to know briefly how the genomic and transcriptomic analyses led to the conclusion that P. antarctica is oleaginous.

3- Lines 198-199: hard to read. it is better to say "metabolism related genes" rather than "metabolism genes"

4- Line 253: glyocylate to glyoxylate.

5- it is strongly recommended to show the expression fold changes of MEL biosynthetic genes and transport in a single separate figure similar to figure 3.

6- English style and grammer needs some improvement for a more easy reading.

Reviewer #4: The results embody this manuscript is a continuation of their previous studies, and has merit. This manuscript can be published with some minor modifications. The suggested changes are as follows:

Full Title: Production characteristics of mannosyl.....

Short Title: Metabolic transcriptomics.....

Abstract L34-36: These results suggest that central metabolism of.......

L85-86: These results ... ....... ...... modification of its central metabolic system.

L386-391: Based on the transcriptomic characterization and biochemical changes - the authors concluding statements are far and wide or out of focus.

6. PLOS authors have the option to publish the peer review history of their article (what does this mean?). If published, this will include your full peer review and any attached files.

Reviewer #1: No

Reviewer #2: No

Reviewer #3: Yes: Ahmad M. Abdel-Mawgoud

Reviewer #4: No

---

## [Author Response · Author response to Decision Letter 0]

1 Oct 2019

We would like to thank you and 4 reviewers for careful reading our manuscript and for giving useful comments. According to the reviewers’ comments, we have revised the manuscript, tables, and figures. We enclosed the revised manuscript and our replies to the comments by the reviewers together with a marked-up manuscript without figure.

We hope that you will consider this revised version suitable for publication in the PLOS ONE.

---

## [Decision Letter · Decision Letter 1]

16 Oct 2019

PONE-D-19-22493R1

Production characteristics of mannosylerythritol lipids in Pseudozyma antarctica T-34 based on gene expression of central metabolic pathways

PLOS ONE

Dear Dr Morita,

Thank you for submitting your manuscript to PLOS ONE. After careful consideration, we feel that it has merit but does not fully meet PLOS ONE's publication criteria as it currently stands. Therefore, we invite you to submit a revised version of the manuscript that addresses the points raised during the reviewing process. We would appreciate receiving your revised manuscript by mid novembre . To enhance the reproducibility of your results, we recommend that if applicable you deposit your laboratory protocols in protocols.io, where a protocol can be assigned its own identifier (DOI) such that it can be cited independently in the future. For instructions see: http://journals.plos.org/plosone/s/submission-guidelines#loc-laboratory-protocols

We look forward to receiving your revised manuscript.

Kind regards,

Marie-Joelle Virolle, PhD

Academic Editor

PLOS ONE

Reviewers' comments:

Reviewer's Responses to Questions

**Comments to the Author**

1. If the authors have adequately addressed your comments raised in a previous round of review and you feel that this manuscript is now acceptable for publication, you may indicate that here to bypass the “Comments to the Author” section, enter your conflict of interest statement in the “Confidential to Editor” section, and submit your "Accept" recommendation.

Reviewer #1: (No Response)

Reviewer #2: (No Response)

Reviewer #3: All comments have been addressed

2. Is the manuscript technically sound, and do the data support the conclusions?

Reviewer #1: Partly

Reviewer #2: Yes

Reviewer #3: Yes

3. Has the statistical analysis been performed appropriately and rigorously? 

Reviewer #1: I Don't Know

Reviewer #2: Yes

Reviewer #3: Yes

4. Have the authors made all data underlying the findings in their manuscript fully available?

Reviewer #1: Yes

Reviewer #2: Yes

Reviewer #3: Yes

5. Is the manuscript presented in an intelligible fashion and written in standard English?

Reviewer #1: Yes

Reviewer #2: Yes

Reviewer #3: Yes

6. Review Comments to the Author

Reviewer #1: Comments on the revice

Most comments were resolved still some aren’t.

1)

There are claims about higher productivities and higher growth but these are not supported with the right data.

MEL production is measured by TLC which is a qualitative measurement technique and not a quantitative one.

I have the feeling that the author cannot or won't quantify there MEL's with an adequate technique (e.g. HPLC-ELSD).

Biomass growth is measured with cell wet weight after MeOH extraction. There are so many biasses on this technique (substrate, lipid extraction by MeOH, evaporation, cell size, lipid content,....).

So I suggest, either you analyse your production parameters (in g/(L*h)). If this is no option than you will have to remove your claims of higher production or state that a TLC analysis based on visual comparison can only give a small hint for higher concentration.

Biomass should be measured with another technique than cell wet weight like e.g. cell dry weight after removal of product by ethyl acetate washing. (don't use MeOH, ethyl acetate will evaporate).

If there is a difference in biomass than should the productivity be linked to the biomass as well (specific productivity). This will probably result in interesting findings.

2)

Figure 2 B&C: As described in your article, “Genome and transcriptome analysis of the asidiomycetous yeast Pseudozyma Antarctica producing extracellular glycolipids, mannosylerythritol lipids”, there were gene transcriptions who were significantly different (p-value <0.005). Can you state that you only used these ones for further analysis when you were using the selection criterium M<0. This will improve your research by confirming that you lowered the false positive rate.

3)

Line 372 & 387: change large amounts to larger amounts or just to “presence of vegetable oil”.

4)

381: precious report should be changed to previous

5)

figure 5 A: the µl of sample should be under the respective lane for clarity.

Reviewer #2: Dears Authors,

Most of my concerns have been addressed in your revised version. But there are still two points which have not been answered satisfactory.

Figure 1: There is still a mistake in the biosynthesis pathway. Please look at Hewald et al. (2006) or Deinzer et al. (2019). Mat1 acetylates C4 and C6 residues of the mannose and not as depicted C2. I propose to cite these articles as basis for your figure. Do you have any hints that Mac2 acylates first then Mac1, or it is still done in a cooperative manner? Please change the figure accordingly to the literature.

The concentration (molarity) of NH4OH-solution is still missing.

Reviewer #3: The manuscript looks much better and is sound for publication in its current status. Yet I suggest the title be changed from :

Production characteristics of mannosylerythritol lipids in Pseudozyma antarctica T-34

based on gene expression of central metabolic pathways

to

Targeted transcriptomic study of the implication of central metabolic pathways in mannosylerythritol lipids biosynthesis in Pseudozyma antarctica T-34

7. PLOS authors have the option to publish the peer review history of their article (what does this mean?). If published, this will include your full peer review and any attached files.

Reviewer #1: No

Reviewer #2: No

Reviewer #3: Yes: Ahmad Abdel-Mawgoud Saleh

---

## [Author Response · Author response to Decision Letter 1]

27 Nov 2019

We would like to thank you and 3 reviewers again for careful reading our manuscript and for giving useful comments. According to the reviewers’ comments, we have revised the manuscript tables and figures. We enclosed the revised manuscript and our replies to the comments by the reviewers together with a marked-up manuscript.

We hope that you will consider this revised version suitable for publication in the PLOS ONE.

---

## [Decision Letter · Decision Letter 2]

17 Dec 2019

Targeted transcriptomic study of the implication of central metabolic pathways in mannosylerythritol lipids biosynthesis in Pseudozyma antarctica T-34

PONE-D-19-22493R2

Dear Dr. Tomotake Morita,

We are pleased to inform you that your manuscript has been judged scientifically suitable for publication and will be formally accepted for publication once it complies with all outstanding technical requirements.

With kind regards,

Marie-Joelle Virolle, PhD

Academic Editor

PLOS ONE

Additional Editor Comments (optional):

Reviewers' comments:

Reviewer's Responses to Questions

**Comments to the Author**

1. If the authors have adequately addressed your comments raised in a previous round of review and you feel that this manuscript is now acceptable for publication, you may indicate that here to bypass the “Comments to the Author” section, enter your conflict of interest statement in the “Confidential to Editor” section, and submit your "Accept" recommendation.

Reviewer #2: (No Response)

2. Is the manuscript technically sound, and do the data support the conclusions?

Reviewer #2: (No Response)

3. Has the statistical analysis been performed appropriately and rigorously? 

Reviewer #2: (No Response)

4. Have the authors made all data underlying the findings in their manuscript fully available?

Reviewer #2: (No Response)

5. Is the manuscript presented in an intelligible fashion and written in standard English?

Reviewer #2: (No Response)

6. Review Comments to the Author

Reviewer #2: All points of my second review have been changed satisfactory. One Proposal: Citation of Hewald et al., 2006, in which the MEL biosynthesis pathway was described.

7. PLOS authors have the option to publish the peer review history of their article (what does this mean?). If published, this will include your full peer review and any attached files.

Reviewer #2: No

---

## [Editor Report · Acceptance letter]

23 Dec 2019

PONE-D-19-22493R2 

Targeted transcriptomic study of the implication of central metabolic pathways in mannosylerythritol lipids biosynthesis in Pseudozyma antarctica T-34 

Dear Dr. Morita:

I am pleased to inform you that your manuscript has been deemed suitable for publication in PLOS ONE. Congratulations! Your manuscript is now with our production department. 

With kind regards,

on behalf of

Dr. Marie-Joelle Virolle 

Academic Editor

PLOS ONE